# In Vivo Biocompatibility Analysis of a Novel Barrier Membrane Based on Bovine Dermis-Derived Collagen for Guided Bone Regeneration (GBR)

**DOI:** 10.3390/membranes12040378

**Published:** 2022-03-30

**Authors:** Carolin Lindner, Said Alkildani, Sanja Stojanovic, Stevo Najman, Ole Jung, Mike Barbeck

**Affiliations:** 1BerlinAnalytix GmbH, 12109 Berlin, Germany; carolin.lindner@berlinanalytix.com (C.L.); saidkildani@gmail.com (S.A.); 2Department for Cell and Tissue Engineering, Faculty of Medicine, University of Niš, 18000 Nis, Serbia; sanja.genetika.nis@gmail.com (S.S.); stevo.najman@gmail.com (S.N.); 3Department of Biology and Human Genetics, Faculty of Medicine, University of Niš, 18000 Nis, Serbia; 4Clinic and Policlinic for Dermatology and Venereology, University Medical Center Rostock, 18057 Rostock, Germany; ole.tiberius.jung@gmail.com

**Keywords:** bovine collagen, resorbable barrier membrane, Guided Bone Regeneration (GBR), biodegradation, biomaterial scoring, DIN EN ISO 10993, immune response, macrophages, transmembraneous vascularization

## Abstract

Collagen-based barrier membranes are nowadays the prevalent option for Guided Bone Regeneration (GBR) procedures. Xenogeneic collagen is highly biocompatible as it shares a similar structure to native human collagen, which prevents it from eliciting an exaggerated host immune response. Most commercially available collagen barrier membranes are porcine-derived, while bovine-derived alternatives are still rarely available. The aim of the present study was to investigate the tissue responses and the barrier functionality of a novel GBR membrane composed of bovine collagen type I (BM). Therefore, the subcutaneous implantation model in Wistar rats was performed to compare the novel medical device with two already clinically used native porcine-based barrier membranes, i.e., Jason^®^ membrane (JM) and Bio-Gide^®^ (BG), at 10-, 30-, 60-, and 90-days post implantationem. Histochemical and immunohistochemical stains were used for histopathological evaluation including a biocompatibility scoring according to the DIN EN ISO 10993-6 norm as well as histomorphometrical analyses of the occurrence of M1 and M2 macrophages and the transmembraneous vascularization. The bovine membrane exhibited a host tissue reaction that was comparable to both control materials, which was verified by the scoring results and the histomorphometrical macrophage measurements. Moreover, the novel membrane exhibited an integration pattern without material fragmentation up to day 60. At day 90, material fragmentation was observable that allowed for “secondary porosity” including transmembrane vascularization. The results of this study suggest that the novel bovine barrier membrane is fully biocompatible and suitable for indications that require GBR as a suitable alternative to porcine-sourced barrier membranes.

## 1. Introduction

As in most medical fields, intensive research into new biocompatible materials is continuously being carried out in the fields of dentistry and oral and maxillofacial surgery. For conduction of the technique of Guided Bone Regeneration (GBR), a procedure that is intended to allow new bone formation within a jaw defect area for later placement of prosthetic dental implants, barrier membranes are designed to prevent non-osteogenic tissue, such as the rapidly proliferating gingival epithelial cells, from growing into the periodontal, peri-implant or alveolar bone defect [1,2,3,4,5]. 

Nowadays, resorbable membranes composed of collagen from various xenogeneic sources are mainly applied in the daily clinical practice [1]. This is due the good biocompatibility of collagen-based biomaterials that has already been proven in case of a broad variety of medical devices such as skin substitutes, or as cardiac or vascular patches [2,6,7,8,9]. Most of the barrier membranes are made from collagen type I and/or III from porcine sources, whereby native collagen won from the dermis or the pericardium have already shown to provide a sufficient barrier functionality [7,10,11]. Membranes with a lifetime of at least two to three months have proven to be most suitable for GBR procedures [12].

Although, such barrier membranes are already available, research in the last decade has elucidated the knowledge on the mechanisms controlling membrane–host interactions in GBR procedures [13]. In this context, Omar et al. published a review article that focused on the possible role of GBR membranes as bioactive compartments in addition to their established role as barriers [13]. The authors stated that barrier membrane contribute bioactively to the osseous regeneration as the cellular and molecular activities towards the membranes are intimately linked to the bone regeneration cascade. They finally concluded that the optimal membrane should play an active role via modulation of molecular activities of the membrane-associated cells during the GBR procedure, in addition to its pure barrier functionality. In this context, it has been shown that the overlying collagen membrane and especially the improvement of its vascularization has a pivotal role for successful bone tissue regeneration [14]. However, a recent in vivo study showed that the available native dermis-derived collagen membranes only allow for a low transmembraneous vascularization in contrast to a membrane based on reconstituted collagen derived from the bovine dermis [15]. This study also showed that the bovine collagen membrane underwent a special integration pattern as their fragments were found to be overlapping starting from day 60 post implantationem, providing a so-called “secondary porosity” in combination with the ingrowth of connective tissue and higher numbers of blood vessels. Although it was proposed in a further in vivo study that the fragmentation of a GBR membrane might interfere with its barrier functionality, this study revealed comparable results of the bovine membrane in terms of biocompatibility and standing time even in comparison to the porcine control group. Thus, it was assumed that the transmembraneous vascularization pattern of this bovine membrane can potentially serve as a further improvement to support the restitutio ad integrum of jawbone defects. This supposition is based on the fact that the implantation bed vascularization of bony defect sides starts usually from the surrounding unaffected bone tissue and grows upwards into the defect [14]. However, transmembraneous vascularization is supposed to allow an entire vascularization pattern—also outgoing from the overlaying flap—that may help for a fast vessel ingrowth into the complete grafting side.

In this context, it has to be mentioned that the tissue response, i.e., the involved cells and the specific manifestation, to a biomaterial in general and thus also to barrier membranes is dependent on the entirety of the specific physicochemical material properties [4]. Factors such as the surface texture of biomaterials have shown to influence processes such as the migration or induction of phagocytes such as macrophages but also multinucleated giant cells (MNGCs) and thus material degradation but also the integration pattern or vascularization and finally tissue regeneration amongst other different factors [6,10,15,16,17,18,19,20,21,22,23]. Material parameters of collagen-based GBR membranes such as tissue source/origin, different manufacturing processes, and post-fabrication processing such as crosslinking have already been identified to influence the integration, as well as the standing time [6,15,16,17,18,19,20,21,22,24,25]. 

Moreover, is has been revealed that the interaction of a resorbable biomaterial (such as a barrier membrane), with cells of the immune system including macrophages, is not only as phagocyting cells but also as key players of the tissue reaction cascade via expression of a broad spectrum of signaling molecules, which is most crucial for the biomaterial’s fate and clinical suitability [4]. Within the last decade it has been elucidated that especially the polarization of macrophages, i.e., their overall expression profile that allows for a functional classifications into pro-inflammatory M1 and anti-inflammatory M2 subtypes, and the overall induction of an inflammatory profile by a biomaterial seems to dictate its respective regenerative fate [26]. Altogether, it is of high importance to test both the tissue compatibility and integration pattern of biomaterials such as barrier membranes via preclinical study models to predict its clinical functionality and the interactions of such materials with the immune system for translation into the clinical application. According to the DIN EN ISO 10993-6 norm, implant devices must be tested in vivo via initial implantation using the subcutaneous implantation model in small animals, which allows for an analysis of the tissue reactivity of a novel medical device via a specialized scoring system and comparison with the reaction towards at least one control device (Figure 1) [27,28]. 

Thereby, other host responses are also given a score, i.e., presence or occurrence of MNGCs, necrosis, fibrosis, tissue degeneration, neovascularization, and fatty infiltration. Furthermore, it has manifoldly been shown that an analysis of the overall immune response based on immunohistochemical detection of macrophage subtypes and their histomorphometrical quantification allows for a more detailed biocompatibility analysis of biomaterials and especially of collagen-based devices [29,30]. Additionally, the analysis of the implantation bed vascularization is a topic of many publications as it is stated to be another important key factor for the successful clinical application of biomaterials [9,31,32,33].

Thus, the present publication intends to investigate the biocompatibility, tissue integration behavior, immune response, and vascularization pattern of a novel bovine dermis-derived membrane as an alternative to commercially available porcine-based barrier membranes that are commonly used in GBR. Histopathological analyses (including the histopathological scoring based on the DIN EN ISO 10993-6 and histomorphometrical analyses of the membrane thickness, the occurrence of M1 and M2 macrophages, and the transmembraneous vascularization were applied based on previously published methodologies [34].

## 2. Materials and Methods

### 2.1. Materials

A newly developed bovine dermis-based collagen membrane (test article) and two commercially available porcine dermis-derived barrier membranes (positive control 1 and 2) were examined in the present study (Table 1). 

#### 2.1.1. Native Dermis-Derived Bovine Membrane

The native bovine dermis-based membrane (Viscofan BioEngineering GmbH, Weinheim, Germany) is mainly composed of type I collagen. A highly standardized proprietary purification process is applied to preserve intact collagen fibers. Membranes extruded from this fiber suspension do not require the addition of any stabilizing chemicals and result in high-purity collagen sheets. The membranes are sterilized with gamma radiation.

#### 2.1.2. Native Dermis-Derived Porcine Membrane

The native porcine dermis-based Bio-Gide^®^ membrane (Geistlich Pharma AG, Wolhusen, Switzerland) is exclusively produced from Swiss porcine tissue and consists of collagen types I and III [35]. The purification process includes an intensive multi-stage chemical purification process and a final sterilization by gamma radiation [36]. This collagen membrane is composed of a compact, smooth layer with low porosity and a porous layer with a sponge-like three-dimensional structure. The Bio-Gide^®^ membrane has already been investigated in many preclinical and clinical studies for its biocompatibility and its usefulness in GBR techniques [29,37].

#### 2.1.3. Native Pericardium-Derived Porcine Membrane 

The native porcine pericardium-derived Jason^®^ membrane (botiss biomaterials GmbH, Zossen, Germany) consists of differentially oriented collagen fibers based on collagen type I and type III that form a comb-like structure and are characterized by strong multidirectional connections [38]. The precursor tissue undergoes a standardized manufacturing and purification process, involving wet chemical treatment followed by freeze-drying. Finally, the biomaterial is sterilized using ethylene oxide gas. This membrane has been tested many times in preclinical and clinical studies [15,29].

### 2.2. In Vivo Experimental Procedure

#### 2.2.1. Experiment Design

Based on a prior power analysis, including an additional drop-out rate of 5% (effect size 1.3, G*Power), a number of six experimental animals per study group and time point (*n* = 6) was used in order to obtain reliable and comparable results. Furthermore, a sham operation group to determine the influence of the operation per se was conducted with the same animal number. Thus, a total of 96 male Wistar rats were included in the present study to analyze the tissue responses to the test materials and both control membranes as well as the sham operation at 10-, 30-, 60-, and 90-days post implantationem. 

The in vivo experiments were performed after the approval of the Local Ethical Committee of the Faculty of Medicine University of Niš based on the decision of the Veterinary Directorate of the Ministry of Agriculture, Forestry and Water Management of the Republic of Serbia.

None of the experimental animals died after the surgical procedure and no abnormalities were observed in the postoperative course. Neither necrosis nor conspicuous signs of inflammation were observed in the animals. No replacement animals were needed to replace unsuitable animals during explantation of the biomaterials.

#### 2.2.2. Animal Husbandry

To allow the experimental animals to acclimate to the new environment, they were brought to the experimental facility 7 days prior to the surgical procedure. During the experiment, the animals were housed in macrolon cages that were sprinkled with softwood granules. The climate in the premises was set at 20–24 °C with medium humidity. In an automatic 12 h rhythm, the room lighting was regulated between light and dark cycles. 

Drinking water was supplied to the animals ad libitum. Food was also provided to the animals without restriction in the form of a standard diet. The cages of the animals were cleaned daily.

The body functions of the experimental animals, such as respiration, body temperature, and mobility, were checked after the surgical procedure at one hour and after three hours to avoid complications after anesthesia and to ensure a smooth awakening. Subsequently, the animals were subjected to daily monitoring with documentation of postoperative condition, general well-being, and wound healing progress.

#### 2.2.3. Subcutaneous Implantation Model According to DIN EN ISO 10993-6

The surgical procedure was performed in the experimental phase. For the surgery, the experimental animals were anesthetized with a combination of ketamine [90 mg/kg] and xylazine [10 mg/kg] by intraperitoneal injection, according to the guidelines for rat anesthesia. The anesthetized animals were then shaved and disinfected in the upper dorsal region, and a transverse incision was made below the scapula region with a scalpel. A subcutaneous pocket in the connective tissue was then prepared bluntly into which the collagen membrane pieces were inserted after an incubation in saline for 10 min. Finally, the incision was closed with a standard suture material (Prolene 6.0, Ethicon, Norderstedt, Germany). During the tapering of the anesthesia, the animals were kept warm on a heating plate (control plate HT 400 W1 with heating plate, Minitube GmbH, Tiefenbach, Germany). The animals were afterwards each placed in a macrolon cage until the first indication of awakening. 

#### 2.2.4. Explantation and Fixation

Immediately following euthanasia of the experimental animals at the respective study time points with an intraperitoneally injected overdose of ketamine and xylazine, explantation of the entire implantation area in combination with the surrounding tissue was performed. Afterwards, the biopsies were placed in a four percent neutral buffered formaldehyde solution for 24 h. 

### 2.3. Histology and Immunohistochemistry

Sample workup was initialized by taking tissue samples from the formalin solution. The explants were then cut into three to four segments of equal size and placed in labeled cassettes (Histosette^®^, VWR, Darmstadt, Germany). In the embedding cassettes, the samples moved through the further workup in a citadel tissue embedder (Automatic Tissue Processor MTP, SLEE medical GmbH, Mainz, Germany) in the next step. This preparation contained an alcohol series in increasing concentrations from 60 to 100% ethanol as well as xylene to dehydrate the tissue samples and finally two paraffin baths to infiltrate the tissue. This step was followed by embedding the tissue specimens in paraffin blocks. To perform the histochemical and immunohistochemical stainings, 3–5 micrometer-thin sections of the biopsies were prepared. The sections were histochemically stained with hematoxylin and eosin (H&E). Furthermore, immunohistochemical detection of integrin alpha x (CD11c) (abx231412, Abbexa Ltd., Milton, UK), hemoglobin scavenger receptor (CD163) (ab182422, abcam, Cambridge, UK), and platelet endothelial cell adhesion molecule (CD31) (ab182981, abcam, Cambridge, UK) were performed as previously described [27,29]. In brief, the slides were incubated with citrate buffer and proteinase K at pH 8 for 20 min in a water bath at 96 °C. Afterwards, an equilibration using TBS-T buffer was conducted and then the tissue was treated via H_2_O_2_ and avidin and biotin blocking solutions (Avidin/Biotin Blocking Kit, Vector Laboratories, Burlingame, CA, USA). Subsequently, the sections were incubated with the respective first antibody and the secondary antibody (goat anti-rabbit IgG-B, sc-2040, 1:200, Santa Cruz Biotechnology, Shandon, CA, USA) for 30 min, respectively. Finally, the avidin–biotin–peroxidase complex (ThermoFisher Scientific, Dreeich, Germany) (30 min) was used followed by a counterstaining via bluing was conducted.

#### 2.3.1. Histopathological Analysis

For the qualitative histopathological analysis, the H&E-stained slides were used. The implant beds were viewed microscopically and the local tissue responses as well as their integration and degradation patterns were assessed. Photographs were taken using an Axiocam 105 color camera that was connected to its software ZEN Core (Zeiss, Oberkochen, Germany).

#### 2.3.2. Histopathological Biomaterial Scoring

The in vivo tissue response to the different barrier membranes was investigated in accordance to the DIN EN ISO 10993-6 scoring system [28]. The local effects were evaluated by a comparison of the tissue response to the test material with that caused by the control materials (positive control 1 and 2), whose clinical acceptability and biocompatibility characteristics have already been proven [28]. Thereby, different cell types and different other tissue responses were evaluated and a respective score value was given based on the value ranges in Table 2. 

Based on the scoring results, the irritancy score was calculated as described in Table 2. and the overall irritancy score of the test article at each study time point was calculated as follows: 

Overall irritancy score = test article irritancy score—average irritancy score of control articles.

If the result was a negative number, the irritancy score was considered to be 0.0.

The irritancy grade was then determined according to Table 3. 

#### 2.3.3. Histomorphometrical Analysis

Quantitative histomorphometrical measurements of the thickness of the membranes using the H&E-stained slides and of the inflammatory tissue response to the biomaterials based on the immunohistochemistry-stained slides were performed as previously described by Barbeck et al. [31]. In brief, the thickness of every membrane was measured at each experimental time point to determine the biodegradation and volume stability of the biomaterials in the animal connective tissue. The analysis was performed according to previously published methods [29]. After digitization of the H&E-stained slides using a special scanning microscope (M8, precipoint, Munich, Germany). The measurements were performed with the “annotations and measurements” tool of the open-source software (ImageJ, U. S. National Institutes of Health, Bethesda, ML, USA). To evaluate the mean membrane thickness at every time point, the thickness of every membrane in each biopsy was measured at 15 different points and the thickness was calculated in mm. 

For the measurement of the numbers of M1 and M2 macrophages in the region of interest (ROI), the immunohistochemically stained slides were digitized, and the cell numbers were counted using an ImageJ-plugin developed by Lindner et al. and related to the respective ROI area (cells/mm^2^) [34]. 

For the histomorphometrical measurements of the vascularization, the CD31-stained sections were initially digitized. The measurement of vascularization on basis of the digitized slides was performed by manual measurement of the parameters of vessel density (vessels/mm^2^), percentage vascularization (percentage of vessel area in the area of the respective implantation site), and the vessel size. The measurement of the vessel density and the percentage vascularization was based on the determination of the areas of the peri-implant implantation beds of the membranes and the respective membrane areas. Subsequently, the individual vessels within these areas were marked. These measurement data further served as a basis for the determination of the vessel density as well as the calculation of the percentage vascularization. The measure of vessel density was obtained from the quotient of the number of vessels and the two areas (number of vessels/mm^2^), whereas the percentage vascularization was calculated from the sum of the vessel areas to the total area of the respective area (∑ vessel areas/total area). Furthermore, the vessel sizes were calculated by the ImageJ software.

### 2.4. Statistical Analysis

Statistical analyses of the quantitative data from the membrane thickness measurements and macrophage counts were performed using an analysis of variance (ANOVA) of the independent sample data. A post hoc test, i.e., a Bonferroni test for Least Significant Difference (LSD), was then performed to determine significant differences between the study groups. For the statistical analyses, the software GraphPad Prism (version 9.0.0, GraphPad Software Inc., La Jolla, CA, USA) was used. The graphs displaying the means with their corresponding standard deviations were also generated via GraphPad Prism. The statistical differences are distinguished into two groups: intraindividual differences (*) or interindividual differences (#). Thereby, the *p*-values were compared with the significance level α and the significances are determined as follows: significant: *p* < 0.05 (#/*), and highly significant: *p* < 0.01 (##/**), *p* < 0.001 (###/***), and *p* < 0.0001 (####/****).

## 3. Results

### 3.1. Histopathology Evaluation

#### 3.1.1. Tissue Response/Inflammation and Tissue Integration

The histopathological evaluation showed at day 10 post implantationem that all membranes were detectable within their subcutaneous implantation beds without histological signs of exaggerated material-induced inflammatory cell or tissue responses (Figure 2A–D). Furthermore, no signs of material fragmentations or material breakdowns were visible.

A mild inflammation was observable within the implant beds of all analyzed membranes at this time point that was comparable to the tissue response due to the surgery (sham operation) (Figure 2A–D and Figure 3A–D). Mainly macrophages were found within the implant beds beside lower numbers of polymorphonuclear cells, lymphocytes, and fibroblasts (Figure 3A–D). Furthermore, low numbers of multinucleated giant cells were found at the material-tissue interfaces of all membranes at this early time point but not within the sham operation areas (Figure 3A–D). The analysis of the cell migration revealed that only single cells were found within the superficial regions of the membranes in the groups of the bovine membrane and the Bio-Gide^®^ membrane (Figure 2A–D and Figure 3A–D). In contrast, the Jason^®^ membrane showed signs of a complete cell invasion at this early study time point (Figure 2A–D and Figure 3A–D).

The histopathological evaluation showed at day 30 post implantationem that all membranes were still detectable within their subcutaneous implantation beds without histological signs of exaggerated material-induced inflammatory cell or tissue responses (Figure 2E–H). No signs of material fragmentation or material breakdowns were observed in the groups of the bovine membrane, the Jason^®^ membrane and the Bio-Gide^®^ membrane (Figure 2E–H and Figure 3E–H). A mild inflammation was still observable within the implant beds of the groups of the bovine membrane, the Jason^®^ membrane and the Bio-Gide^®^ membrane (Figure 2E–H and Figure 3E–H). Furthermore, the tissue reaction pattern in these groups were comparable to the tissue responses in the sham operation group (Figure 2E–H and Figure 3E–H). Thereby, at day 30 post implantationem mainly macrophages and lower numbers of polymorphonuclear cells, lymphocytes, and fibroblasts were involved in the tissue reactions to these membranes (Figure 3E–H). Additionally, minor numbers of multinucleated giant cells (MNGCs) were found within the implant beds of all three membranes. The analysis of the cell migration revealed that only single cells penetrated the superficial regions of the bovine membranes at this time point, while the material bodies of the Jason^®^ membranes and the Bio-Gide^®^ membranes were completely penetrated mainly by macrophages (Figure 2E–H).

At day 60 post implantationem, the histopathological evaluation showed that the membranes in the groups of the bovine membrane, the Jason^®^ membrane, and the Bio-Gide^®^ membrane were still detectable within their subcutaneous implantation beds without histological signs of exaggerated material-induced inflammatory cell or tissue responses (Figure 2I–L). Still no signs of material fragmentation or material breakdowns were observed in the groups of the Jason^®^ membrane and the Bio-Gide^®^ membrane (Figure 2I–L and Figure 3I–L). In contrast, the materials in the group of the bovine membrane showed initial signs of material breakdowns as in some spots an ingrowth of cells and complex tissue were observable (Figure 2I). However, in none of the biopsies from this group a complete material fragmentation was observed. A mild inflammation was still observable within the implant beds of the groups of the Jason^®^ membrane and the Bio-Gide^®^ membrane (Figure 2I–L and Figure 3I–L) that was comparable to the tissue responses in the sham operation group (Figure 2I–L and Figure 3I–L). In both these groups mainly macrophages and low numbers of polymorphonuclear cells, lymphocytes, and fibroblasts were involved in the tissue reactions (Figure 3I–L). In the group of the bovine membrane also a very mild inflammation was found within those regions that showed no signs of a material breakdown also mainly including macrophages and low numbers of polymorphonuclear cells, lymphocytes, and fibroblasts. Within the breakdown spots a slightly higher inflammatory tissue response was detectable indicated mainly by slightly higher numbers of multinucleated giant cells (Figure 2I–L and Figure 3I–L). However, the overall tissue responses in this group were also comparable to the observation in the groups of the Jason^®^ membrane, the Bio-Gide^®^ membrane, and the sham operation group. The analysis of the cell migration revealed that only single cells penetrated the superficial regions of the bovine membranes within those regions that showed no material breakdowns at this time point, while a higher cell penetration was found within the breakdown regions (Figure 3I). Interestingly, the Jason^®^ membranes and the Bio-Gide^®^ membranes were completely infiltrated mainly by macrophages. Lower numbers of inflammatory cells were found within the membrane bodies of JM and GB at this time point compared to the former study time points (Figure 2I–L).

At day 90 post implantationem, the histopathological evaluation showed that the membranes in the groups of the bovine membrane were completely fragmented and histological signs of a slightly higher inflammatory response in comparison to the control materials were observed (Figure 2M–P and Figure 3M–P). In contrast, the membranes in the groups of the Jason^®^ membrane and the Bio-Gide^®^ membrane were still noticeable within their subcutaneous implantation beds without low histological signs of material-induced inflammatory cell or tissue responses (Figure 3M–P). Still no signs of material fragmentation or material breakdowns were observed in the groups of the Jason^®^ membrane and the Bio-Gide^®^ membrane (Figure 2N,O and Figure 3N,O). A mild inflammation was still observable within the implant beds of the groups of the Jason^®^ membrane and the Bio-Gide^®^ membrane (Figure 2N,O and Figure 3N,O) being still comparable to the tissue responses in the sham operation group (Figure 2P and Figure 3P). In both these groups, mainly macrophages and low numbers of polymorphonuclear cells, lymphocytes, and fibroblasts were found (Figure 3N,O). In the group of the bovine membrane, a slightly higher inflammation was found also mainly including macrophages and lymphocytes in combination with higher numbers of multinucleated giant cells (Figure 3M). Additionally, low numbers of polymorphonuclear cells, lymphocytes, and fibroblasts were detectable. Furthermore, the connective tissue that was observable within the interspaces of the material fragments contained high numbers of small and medium-sized vessels. The analysis of the cell migration revealed that high numbers of cells penetrated the material bodies of the bovine membrane at this time point (Figure 2M and Figure 3M). Interestingly, the Bio-Gide^®^ membranes were still completely penetrated mainly by macrophages, while lower numbers of cells were found within the membrane bodies of the Jason^®^ membrane at this time point compared to the former study time points (Figure 2N,O and Figure 3N,O).

#### 3.1.2. Biomaterial Scoring Results and Irritancy Score Calculation

Histopathological scoring of the three barrier membranes and the sham operation resulted in comparable degrees of inflammation at day 10 post implantationem (Table 4). The inflammatory tissue response to the membranes and the sham wound included cells of the immune system, i.e., polymorphonuclear cells/granulocytes (rare presence, bovine-derived membrane exhibited a slight increase), lymphocytes (rare presence, comparable in all groups), plasma cells (rare presence, in sham operation group only), macrophages (moderate presence in the group of the bovine-derived membrane, rare presence in other groups), giant cells (rare presence, comparable in all membrane groups), neovascularization (minimal, comparable in all groups), fibrosis (not detectable, insignificant appearance in the groups of the Bio-Gide^®^ and Jason^®^ membranes), fatty infiltrate (not detectable in all groups), and necrosis (not detectable in all groups). 

Histopathological scoring of the three barrier membranes and the sham operation resulted in comparable degrees of inflammation at day 30 post implantationem (Table 5). The inflammatory tissue response to the membranes and the sham wound included cells of the immune system, i.e., polymorphonuclear cells/granulocytes (rare presence, comparable in all groups), lymphocytes (rare presence, comparable in all groups), plasma cells (rare presence, in sham operation group only), macrophages (moderate presence, comparable in all groups), giant cells (rare presence, comparable between the groups of the bovine-based membrane and the Bio-Gide^®^ membrane, insignificant in the groups of the Jason^®^ membrane and sham operation), neovascularization (minimal, increased in the sham operation group), fibrosis (slightly detectable in the Bio-Gide^®^ membrane group), fatty infiltrate (detectable in the sham operation group), and necrosis (not detectable in all groups). 

Histopathological scoring of the three barrier membranes and the sham operation resulted in comparable degrees of inflammation at day 60 post implantationem (Table 6). The inflammatory tissue response to the membranes and the sham wound included cells of the immune system, i.e., polymorphonuclear cells/granulocytes (rare presence, comparable in all groups), lymphocytes (rare presence, comparable in all groups), plasma cells (not detectable in all groups), macrophages (moderate presence in the group of the bovine-derived membrane, rare presence in other groups), giant cells (rare presence in the group of the bovine-derived membrane, insignificant presence in other membrane groups), neovascularization (minimal in the group of the bovine-derived membrane and the sham operation group), fibrosis (slightly detectable in the groups of the bovine-derived membrane and Bio-Gide^®^ membrane), fatty infiltrate (detectable in in the groups of the bovine-derived membrane and the sham operation group), and necrosis (not detectable in all groups). 

At day 90 post implantationem, the histopathological scoring of the three barrier membranes and the sham operation resulted in comparable degrees of inflammation between the bovine-derived membrane and the Bio-Gide^®^ membrane, and decreased degrees in case of the Jason^®^ membrane and the sham operation group (Table 7). The inflammatory tissue response to the membranes and the sham wound included cells of the immune system, i.e., polymorphonuclear cells/granulocytes (rare presence, comparable in all groups), lymphocytes (rare presence, comparable in all groups), plasma cells (not detectable in all groups), macrophages (moderate presence all membrane groups, rare presence in the other groups), giant cells (rare presence in the group of the bovine-derived membrane, insignificant presence in the other membrane groups), neovascularization (minimal in the group of the bovine-derived membrane and the sham operation group), fibrosis (slightly detectable in the group of the Bio-Gide^®^ membrane), fatty infiltrate (detectable in the sham operation group), and necrosis (not detectable in all groups). 

The irritancy score was calculated based on the scoring results (Table 8). The calculation showed that bovine-derived membrane had an average treatment irritancy of 9.5 at day 10 post implantationem, and an overall irritancy score of 2.9; hence, the biomaterial was considered non-irritant at this time point. At day 30 post implantationem, the treatment irritancy score of the bovine-derived membrane had a total of 7.5, and an overall irritancy score of 0.0; hence, the biomaterial was considered non-irritant at this time point. At 60 days post implantationem, the bovine-derived membrane had an average of 7.88, and an overall irritancy score of 2.84; hence, the membrane was considered non-irritant. Finally, at day 90 post implantationem, the average irritancy score of the bovine-derived membrane was 8.5, and overall irritancy score was 2.85; hence, the biomaterial was considered non-irritant. 

### 3.2. Histomorphometrical Results

#### 3.2.1. Thickness Measurements

The analysis of the thickness measurements showed that differences in thicknesses were detected only at day 90 post implantationem. At this time point, the bovine membrane recorded a significantly higher thickness compared with the control groups (Jason^®^ membrane and Bio-Gide^®^ membrane), (# *p* < 0.05 and #### *p* < 0.0001, respectively) (Table 9 and Figure 4). 

In case for the bovine membrane, a highly significant increase was recorded between the first time points (**** *p* < 0.0001) (Table 9 and Figure 4). Afterwards, the thickness of the membrane decreased with high significance after each time point (**** *p* < 0.0001). Additionally, in the group of the Jason^®^ membrane, there was a significant increase at day 30 (** *p* < 0.01) but no significant decrease was recorded afterwards. The Bio-Gide^®^ membrane exhibited a significant increase at day 30 (****: *p* < 0.0001) and the thickness remained comparable afterwards (Table 9 and Figure 4). 

#### 3.2.2. Immune Response Measurements

The analysis of the occurrence of M1 and M2 macrophages showed at day 10 post implantationem that significantly higher numbers of anti-inflammatory CD163-positive macrophages were found in the study group of the Bio-Gide^®^ membrane group compared to the values in the sham operation group (* *p* < 0.05) (Figure 5, Figure 6 and Table 10). 

No other significant differences were found comparing the numbers of anti-inflammatory macrophages in the different study groups at this time point. Furthermore, the analysis revealed that comparable numbers of pro-inflammatory CD11c-positive macrophages were detectable in all study groups (Figure 5, Figure 6 and Table 10). Moreover, the values of the anti-inflammatory macrophages were significantly higher (### *p* < 0.001, ## *p* < 0.01 and # *p* < 0.05, respective to the Bio-Gide^®^ membrane group, the Jason^®^ membrane group and the bovine membrane group) compared to the values of the pro-inflammatory macrophages in all membrane groups) (Figure 5, Figure 6 and Table 10). 

The analysis of the occurrence of M1 and M2 macrophages also showed at day 30 post implantationem that comparable numbers of anti-inflammatory CD163-positive macrophages were found in all study groups (Table 10, Figure 7 and Figure 8). Furthermore, the analysis revealed that comparable numbers of pro-inflammatory CD11c-positive macrophages were detectable in all study groups (Table 10, Figure 7 and Figure 8). Moreover, the values of the anti-inflammatory macrophages were significantly higher (#### *p* < 0.0001 and ### *p* < 0.001) compared to the values of the pro-inflammatory macrophages in all study groups (Table 10, Figure 7 and Figure 8).

The analysis of the occurrence of M1 and M2 macrophages only showed at day 60 post implantationem that significantly higher numbers of anti-inflammatory CD163-positive macrophages were found in the study group of the bovine membrane compared to the values in the Bio-Gide^®^ membrane group (*** *p* < 0.001) as well as significantly higher numbers in the study group of the sham operation group compared to the values in the Jason^®^ membrane group (** *p* < 0.01) and the Bio-Gide^®^ membrane group (**** *p* < 0.0001) (Table 10, Figure 9 and Figure 10). No other significant differences were found comparing the numbers of anti-inflammatory macrophages in the different study groups at this time point. Furthermore, the analysis revealed that comparable numbers of pro-inflammatory CD11c-positive macrophages were detectable in all study groups (Table 10, Figure 9 and Figure 10). Moreover, the values of the anti-inflammatory macrophages were significantly higher (#### *p* < 0.0001 and ### *p* < 0.001) compared to the values of the pro-inflammatory macrophages in all study groups (Table 10, Figure 9 and Figure 10).

The analysis of the occurrence of M1 and M2 macrophages also showed at day 90 post implantationem that comparable numbers of anti-inflammatory CD163-positive macrophages were found in all study groups (Table 10, Figure 11 and Figure 12). Furthermore, the analysis revealed that comparable numbers of pro-inflammatory CD11c-positive macrophages were detectable in all study groups (Table 10, Figure 11 and Figure 12). Moreover, the values of the anti-inflammatory macrophages were significantly higher (### *p* < 0.001 and (##: *p* < 0.01) compared to the values of the pro-inflammatory macrophages in all study groups (Table 10, Figure 11 and Figure 12).

#### 3.2.3. Vascularization Measurements

The analysis showed that no blood vessels were found within the membrane area in any of the study groups at day 10 post implantationem so that no measurement of the transmembraneous vascularization was conducted at this early time point (data not shown).

The analysis of the vascularization revealed that comparably high numbers of blood vessels were found within the peri-implant tissue of all analyzed membranes at day 30 post implantationem (Figure 13). At this time point, only minimal ingrowth of blood vessels into the membrane area was found in all study groups (Figure 13).

The analysis of the vessel density measurements within the membrane areas showed that comparable values were detected at day 30 post implantationem (Figure 13 and Table 11). 

At days 60 and 90 post implantationem, the histopathological evaluation showed that all membranes were detectable within their subcutaneous implantation beds. In the group of the native bovine membrane a fragmentation was visible combined with an ingrowth of a cell- and vessel-rich connective tissue, while no signs of fragmentation were found in the groups of the Bio-Gide^®^ and the Jason^®^ membrane (Figure 14 and Figure 15).

The analysis of the vascularization revealed that significantly higher numbers of blood vessels were detectable within the peri-implant tissue and within the materials bodies of the native bovine membranes, while only low numbers of vessels were found in both compartments in the groups of the Bio-Gide^®^ and the Jason^®^ membrane (Figure 16 and Table 11).

At day 60 post implantationem, significantly higher numbers of vessels per mm^2^ were found in the group of the native bovine membrane compared to the values in the other study groups (Figure 16 and Table 11). In the groups of the Bio-Gide^®^ membrane and the Jason^®^ membrane group, comparable values were found. 

At day 60 post implantationem, a significantly higher percent vascularization was found in the group of the native bovine membrane compared to the values in the other study groups (Figure 17 and Table 11). In the groups of the Bio-Gide^®^ membrane and the Jason^®^ membrane, comparable values were found. 

At day 60 post implantationem, a significantly higher mean vessel diameter was found in the group of the native bovine membrane compared to the values in the Jason^®^ membrane group (Figure 18 and Table 11). In the group of the Bio-Gide^®^ membrane, the vessel diameter did not significantly differ from the values in the other study groups. 

At day 90 post implantationem, significantly higher vessel numbers detected in the group of the native bovine membrane compared to the values in the other study groups (** *p* < 0.01 and *** *p* < 0.001) (Figure 16 and Table 11). The values in the groups of the Bio-Gide^®^ membrane group and the Jason^®^ membrane group did not significantly differ.

At day 90 post implantationem, a significantly higher percent vascularization was detected in the group of the native bovine membrane compared to the values in the other study groups (Figure 17 and Table 11). The values in the groups of the Bio-Gide^®^ membrane group and the Jason^®^ membrane group did not significantly differ.

At day 90 post implantationem, comparable mean vessel diameter values were detected in all study groups (Figure 18 and Table 11). 

## 4. Discussion

Although porcine-derived collagen membranes have become the new gold standard as resorbable barrier membranes for GBR, recent results have shown that bovine-sourced materials seem to provide a special integration pattern including a so-called “secondary porosity” in combination with a transmembraneous vascularization pattern that can potentially serve as a further basis to support the restitutio ad integrum of jawbone defects [39,40]. Thus, such a barrier membrane may bioactively contribute to the osseous regeneration via generation of a healing-supportive microenvironment during the GBR procedure in addition to its pure barrier functionality. Thus, the present paper aimed to analyze the biocompatibility, tissue integration behavior, immune response, and vascularization pattern of a newly developed bovine dermis-derived collagen membrane. Thereby, two commercially available barrier membranes (Jason^®^ membrane and Bio-Gide^®^ membrane) based on porcine origin tissues served as control materials and sham operations were included to investigate the tissue responses to the implantation procedure per se. Thereby, a qualitative histopathological analysis and quantitative analysis including a scoring according to the DIN EN ISO 10993-6 as well as histomorphometrical analysis of the membrane thickness, the occurrence of M1 and M2 macrophages and the transmembraneous vascularization were carried out based on previously published methodologies [34].

Initially, the results of the histopathological analysis revealed that the novel barrier membrane remained intact until day 60 post implantationem, where slight fragmentation started to appear and complete fragmentation was observed at day 90, along with the infiltration of connective tissue and high vessel numbers. Morphologically, the bovine membrane appeared to have no porosity until day 60 post implantationem, while the interspaces between the material fragments at day 90 allowed for a “secondary porosity” as already described in the case of another study on a bovine-derived membrane [15]. In contrast to a previously analyzed membrane that was completely fragmented already at day 60 post implantationem, the material examined in the present study was shown to be intact until day 90. This time frame between 60 and 90 days may be of vital importance in the context of GBR treatment as it has been figured out that membranes, which provide a barrier functionality of at least two to three months, are most suitable [12]. In this context, it has to be mentioned that two key factors have to be ensured by the application of a dental barrier membrane:(i)A displacement of the underlying bone substitute (granules) has to be prevented for successful bone tissue regeneration [41];(ii)The gingiva has to be hindered from growing into the bone augmentation site to ensure that it does not interfere with the bone tissue regeneration process [42].

Thus, a dental membrane must function as a barrier until (a) the bone substitute material is sufficiently stabilized and (b) the gingiva has healed so that it cannot enter the grafting site and disturb the progress of bone regeneration and wound healing.

In view of the primary stabilization of the bone graft, different studies have revealed that bone substitute materials are integrated within stable connective tissue, including a dimensionally stable extracellular matrix, between day 10 and day 30 after implantation [27,43]. It is assumable that this initial tissue integration provides a primary stability, so that the bone substitute is “fixated” within its implantation bed [44]. Gradually, the stability of the bone graft increases through bone tissue integration via osseoconduction. Complete bony integration results in increased stability but requires several months (minimally 3 months) for the bone substitute integration to reach its maximum strength. Altogether, the critical time period when the graft needs to be supported by a collagen membrane has been identified to be between day 10 and day 30 after implantation [27,43]. Thus, both bovine membranes fulfill this requirement.

In view of the gingival healing, it has to be mentioned that initial (epithelial) wound healing primarily depends on the size of the wound and whether the injury heals through a primary or secondary intention [45]. However, it has been assumed that this phase ends between 2 and 3 weeks [46,47]. The normal response to an initial injury involves three overlapping, however, distinct stages: inflammation, new tissue formation, and remodeling and, moreover, distinct cell populations are involved in these phases of tissue repair [47]. During these events, cells undergo changes in gene expression, most of them driven by cell–matrix interactions and/or initiated soluble mediators [48]. Based on different in vivo study results using both a subcutaneous implantation model in rats and mice and a calvaria implantation model in rats analyzing the tissue reaction to bone substitute materials (covered via collagen membranes), tissue healing of the overlying epithelial layer was observed between day 15 and 30 after implantation [49,50,51,52,53,54,55]. This healing process is approximately comparable to the gingival healing within the oral cavity after bone augmentation procedures. Based on these observations it can be assumed that the minimally required time span for barrier functionality of a GBR membrane can be set to 30 days. This assumption is substantiated by the aforementioned fast resorption times of a few weeks in case of natural collagen membranes as it has already been revealed that the clinical application of this membrane type leads to the desired clinical outcome [56]. Altogether, it can thus be concluded that the novel material analyzed in this study can cover this initial important time period and support the GBR procedure as the observed time period until fragmentation of the membrane includes minimally 60 to 90 days post implantationem.

Moreover, the histopathological analysis showed that the bovine membrane induced a slightly increased inflammatory tissue reaction involving a granulation tissue, whose appearance was similar to that observed in the control groups up to day 60 post implantationem. Multinucleated giant cells (MNGCs) and blood vessels remained on the surface of all biomaterials; however, MNGCs appeared to spread onto the surfaces of the bovine membrane’s fragments starting from day 60 after implantation. This observation indicates that the material fragmentation started from that time point onwards as MNGCs are correlates of the material phagocytosis as already shown in case of different other biomaterials such as bone substitutes [57,58]. Noticeably, the granulation tissue did not decrease in the group of bovine membrane at this late study time point in contrast to the control groups whose (inflammatory) tissue reaction could be described as “passivated”. Thus, this native bovine membrane underwent a different integration pattern compared to the control membranes based on the induction of a higher number of phagocyting cells, which explains the aforementioned integration behavior. The reason for the phagocyte occurrence in this group could not be analyzed in the present study and should thus be part of further studies including material characterization methods. A reason for the different phagocyte migration might be the different “natural” cross-linking degree of the bovine collagen in comparison to the both porcine-based control membranes [59,60,61,62]. In this context, a study conducted by McClain et al. showed that both ovine and porcine collagen seem to have a much higher degree of intermolecular cross-linking than bovine collagen as shown by lower acid-soluble tropocollagen values [59]. Moreover, bovine collagen showed a significantly lower β_12_/β_11_ ratio than ovine or porcine collagen. Furthermore, a higher dimer content of the porcine samples was measured in the β_12_ fraction and its thermal shrinkage temperature was also significantly lower. These data show that the animal source—although collagen is described as a very conservative molecule—and related physicochemical differences might be another reason for the different tissue reactions. However, further (molecular-) biological studies have to clarify these important scientific issues particularly in the context of the further development of biomaterials.

The immunohistochemical detection of the macrophage subtypes showed histologically that pro-inflammatory cells were initially found on the surfaces of all membranes and this subtype expanded to the surface of the fragmented bovine membrane and slightly infiltrated the control groups at day 90 post implantationem. This result is in line with previous observations that concluded that pro-inflammatory cells were mainly involved in biodegradation processes of biomaterials [29,43]. Thus, the observed M1 macrophages found on the surface of the bovine membrane at day 90 suggest that the resorption is still ongoing. Interestingly, most macrophages within the surrounding connective tissue in the early phase after implantation and most infiltrated macrophages in the groups of the control membranes at later time points were of the M2 phenotype. This observation—in combination with the aforementioned “passivation” of the tissue reactions in these study groups—suggests that the bioresorption has slowed down and the membranes were rather integrated than transformed into local tissue. Interestingly, the connective tissue surrounding the fragments of the bovine membrane contained a higher number of M2 macrophages at day 90 post implantationem, which indicates that the bovine membrane promotes tissue healing comparable to the porcine materials at later time points, while being simultaneously resorbed.

Despite the described differences in the tissue reactions, both the DIN EN ISO -based scoring as well as the histomorphometrical analysis of the occurrence of M1 and M2 macrophages, revealed that the newly developed membrane induced an overall tissue response comparable to both control materials which have manifoldly shown to be biocompatible [29,63]. The measurement results altogether showed that the anti-inflammatory cells in total were significantly higher than the numbers of pro-inflammatory cells in all materials groups and also comparable with the sham operation. This result clearly showed that even though the bovine membrane does not experience passivation such as the control membranes, it still promotes a comparable overall tissue reaction and wound healing pattern. Interestingly, the majority of the inflammatory cells were macrophages that increased the scoring values. However, the scoring system does not allow differentiation between pro- and anti-inflammatory subtypes of the individual cell types and therefore also not of the macrophages. Thus, the immunohistochemical detection of the macrophage subpopulations and their occurrence based on histomorphometry was conducted, which showed that most of the macrophages were of the anti-inflammatory type as discussed in more detail below.

Additionally, the described increase in the phagocyte numbers, i.e., especially of MNGCs that were noted on the surface of the bovine membrane and its fragments, is a further explanation for the vascularization pattern of the bovine barrier membrane. The results of the analysis of the membrane vascularization showed that no significant differences of all measured parameters were found comparing the values of the three study groups at day 30 post implantationem. Starting with the fragmentation of the bovine membrane at day 60 post implantationem that was accompanied by the ingrowth of connective tissue and vessels, the vascularization of the membrane area was statistically higher compared to the respective values in the groups of the control membranes as also shown at day 90 post implantationem. Thereby, both the vessel density and also the percent vascularization were higher compared to the values in both control groups. In contrast, the analysis of the mean vessel diameters showed that only at day 60 post implantationem a significant increase in this measurement parameter in the group of the bovine membrane compared to the values in the Jason membrane group was found. These data reveal that the special integration behavior of the bovine membrane induces a significantly higher transmembraneous vascularization that mainly seems to be based on an increase in the vessel density within the ingrowing connective tissue rather than a vessel maturation. In this context, MNGCs have been reported multiple times to increase the implant site vascularization through the expression of mediators such as the Vascular Endothelial Growth Factor (VEGF), which shows again a strong connection between the occurrence of phagocytes and osseo-stimulating approaches such as the vascularization [64]. These results propose that the novel membrane and its late fragmentation, allows for transmembrane vascularization that can finally support the bone regeneration process, as mentioned before. In the case of using this membrane as a barrier membrane in GBR, the bone growth might potentially happen from the implantation bed outwards, as well as from the soft tissue margin inwards. Interestingly, this special integration behavior is not expected with the application of most of the commercially available barrier membranes, while recent discussion in this field promotes a possible role of GBR membranes as bioactive compartments in addition to their established role as barrier materials [9]. In this context, Omar et al. concluded that the optimal membrane should play an active role via modulation of molecular activities of the membrane-associated cells during the GBR procedure in addition to its pure barrier functionality. In this context, it has been shown that the overlying collagen membrane and especially the improvement of its vascularization has a pivotal role for successful bone tissue regeneration [14]. Thus, it can be assumed that the observed vascularization pattern of the bovine membrane can have a positive influence on tissue healing in the context of Guided Bone Regeneration (GBR). 

Finally, the measurement results revealed that the thickness of the three analyzed membranes differed only significantly at day 90 post implantationem, which seems to be related to both variances in the cell migration behavior. In the groups of the newly developed bovine collagen membrane and the pericardium-based porcine membrane, an initial significant increase in the material thickness between day 10 and 30 post implantationem was found, followed by a stepwise significant decrease until the end of the observation period at day 90 post implantationem only in the group of the bovine membrane. However, different mechanisms led to the comparable swelling behavior. In the case of the non-porous novel bovine membrane, the initial swelling seems to be based on water or fluid absorption as no cell ingrowth was observed at this time point. In contrast, a high initial cell migration was found in the group of the porcine pericardium-derived membrane, also shown by a recent study from Ottenbacher et al. that developed a novel histomorphometrical approach that enables to detect such migration differences in case of barrier membranes [65]. Only in the group of the porcine dermis-derived collagen membrane a significant decrease in the thickness between the first two time points was found and no further thickness changes were measured afterwards until day 90 post implantationem as also found in the group of the pericardium-based porcine membrane, which is also in line with the previous study results that showed no significant cell migration into this membrane in this study period. The thickness decrease in the bovine collagen-based membrane at these later time points can be explained by the ongoing degradation and fragmentation and this result thus fits the described histological results.

Thus, only the bovine membrane underwent a stepwise decrease in thickness until the end of the observation period, while the in vivo thickness of both porcine-derived membranes did not change in this time frame. This measurement result is also in agreement with the described histological results, which described a “passivation” of the tissue reaction. Altogether, the results of the thickness measurements revealed significant differences that have no influence on the barrier functionality or other related functionalities and should not lead to (bone) tissue healing issues (e.g., dehiscence). For more clarification, especially of the cell migration into collagen membranes such as the novel bovine membrane, the approach developed by Ottenbacher et al. can further be applied to reveal if the swelling ratios were based on different migration patterns.

Altogether, the results of the present study revealed that the new bovine membrane was integrated within the subcutaneous connective tissue without exaggerated material-induced inflammatory cell or tissue responses. The membrane remained intact until day 60 post implantationem leading to the conclusion that the over- and the underlying compartments were separated from each other minimally until that time point. Complete fragmentation was only observed at day 90, while the fragments lay on top of each other in a roof tile-like arrangement. This special degradation pattern also leads to the conclusion that at later time points the barrier functionality can be maintained. At this time point the epithelial layer usually is already healed, consequently no ingrowth of epithelium should be possible, which leads to the conclusion that the membrane will provide an equivalent barrier functionality towards overlying gingiva when applied at the jawbone.

The bovine membrane exhibited a host tissue reaction that is closely comparable to the control groups, which was verified by the macrophage polarization. The membrane, however, exhibited a different integration pattern including a transmembrane vascularization. The results of this study suggest that this native (non-chemically crosslinked) bovine membrane is biocompatible and is suitable for GBR indications. Thus, the novel bovine membrane can support the bone regeneration process. 

## Figures and Tables

**Figure 1 membranes-12-00378-f001:**
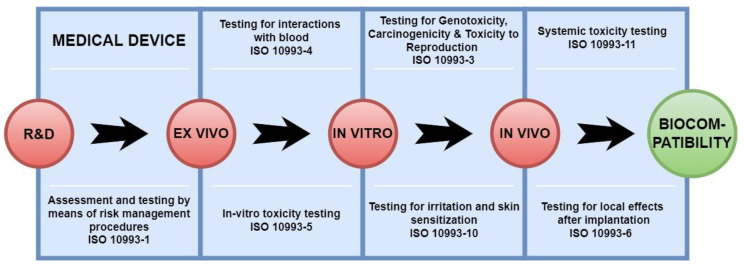
Schematic overview of the development process of medical devices.

**Figure 2 membranes-12-00378-f002:**
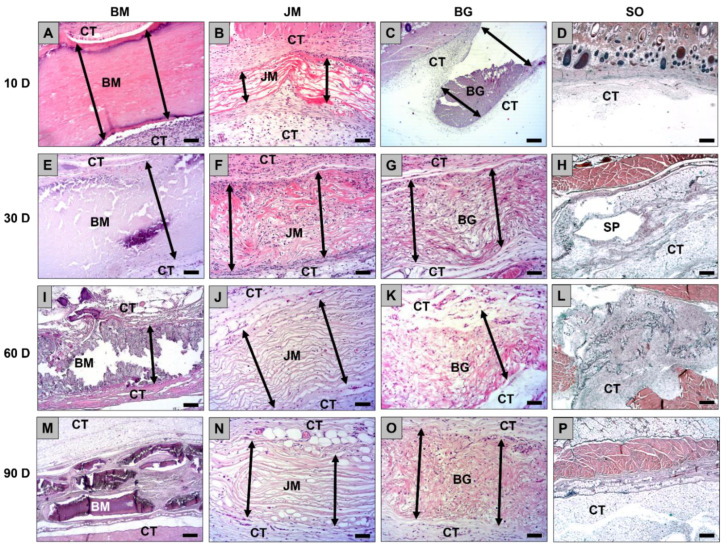
Exemplary histological images of the three membranes (**A**–**C**,**E**–**G**,**I**–**K**,**M**–**O**, double arrows) and the sham operation (**D**,**H**,**L**,**P**) within the subcutaneous connective tissue (CT) at day 10, 30, 60, and 90 post implantationem. SP = subcutaneous pocket of the sham operation. BM = bovine membrane, JM = Jason^®^ membrane, BG = Bio-Gide^®^ membrane, SO = sham operation (H&E-stainings, 40× magnifications, scalebars = 100 µm).

**Figure 3 membranes-12-00378-f003:**
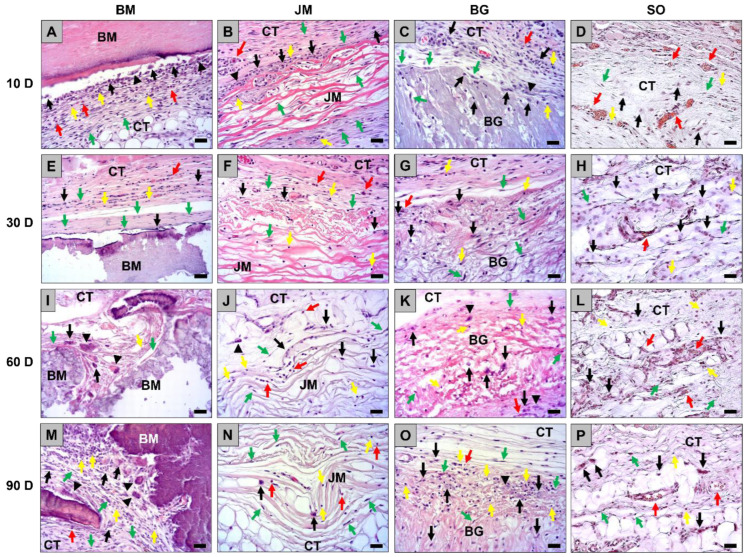
Exemplary histological images of the tissue responses to the three membranes (**A**–**C**,**E**–**G**,**I**–**K**,**M**–**O**) and the sham operation (**D**,**H**,**L**,**P**) within the subcutaneous connective tissue (CT) at day 10, 30, 60, and 90 post implantationem. Black arrows = macrophages, black arrowheads = multinucleated giant cells, green arrows = fibroblasts, red arrows = eosinophilic granulocytes, yellow arrows = lymphocytes. BM = bovine membrane, JM = Jason^®^ membrane, BG = Bio-Gide^®^ membrane, SO = sham operation (H&E-stainings, 200× magnifications, scalebars = 20 µm).

**Figure 4 membranes-12-00378-f004:**
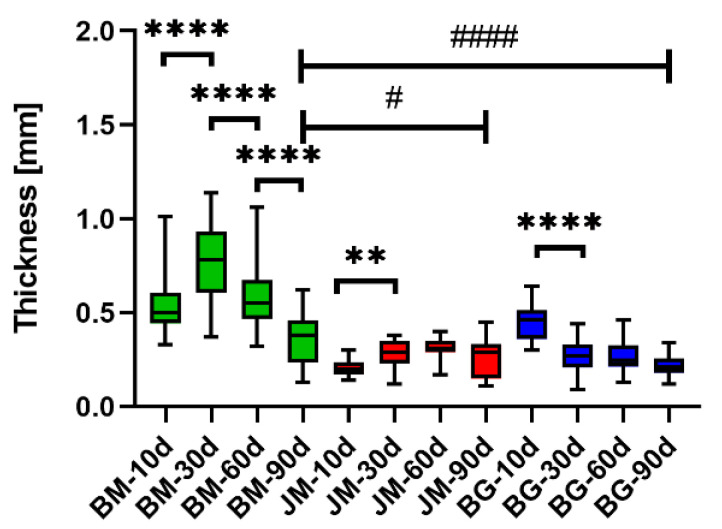
Results of the histomorphometrical measurements of the membrane thickness (* = intraindividual differences, # = interindividual differences; # *p* < 0.05, ** *p* < 0.01 and ****/#### *p* < 0.0001).

**Figure 5 membranes-12-00378-f005:**
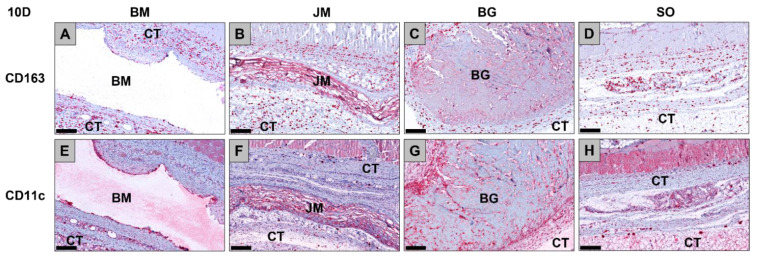
Exemplary histological images of the distribution of anti-inflammatory CD163-positive macrophages (**A**–**D**) and pro-inflammatory CD11c-positive macrophages (**E**–**H**) (red staining) within the implant beds of the three membranes within the subcutaneous connective tissue (CT) at day 10 post implantationem. (**A**) BM = native bovine membrane, (**B**) JM = Jason^®^ membrane, (**C**) BG = Bio-Gide^®^ membrane, (**D**) Sham operation group without biomaterial insertion (CD163-immunostainings (**A**–**D**) and CD11c-immunostainings (**E**–**H**), 200× magnifications, scalebars = 200 µm).

**Figure 6 membranes-12-00378-f006:**
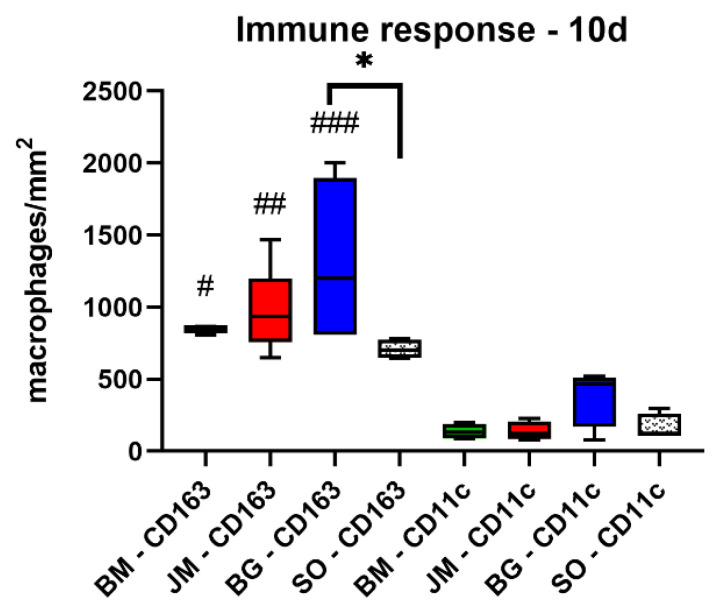
Results of the histomorphometrical measurements of the macrophage distribution at day 10 post implantationem (* = intraindividual differences, # = interindividual differences; ###: *p* < 0.001 and ##: *p* < 0.01, */#: *p* < 0.05).

**Figure 7 membranes-12-00378-f007:**
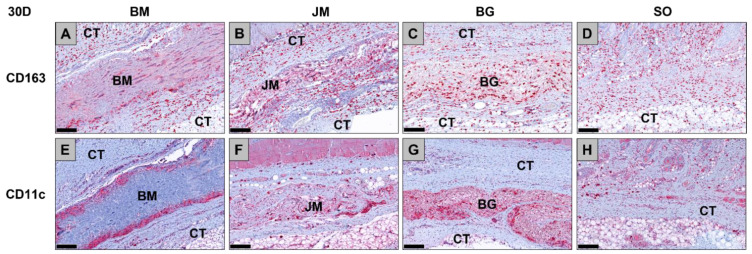
Exemplary histological images of the distribution of anti-inflammatory CD163-positive macrophages (**A**–**D**) and pro-inflammatory CD11c-positive macrophages (**E**–**H**) (red staining) within the implant beds of the three membranes within the subcutaneous connective tissue (CT) at day 30 post implantationem. (**A**) BM = Bovine membrane, (**B**) JM = Jason^®^ membrane, (**C**) BG = Bio-Gide^®^ membrane, (**D**) Sham operation group without biomaterial insertion (CD163-immunostainings (**A**–**D**) and CD11c-immunostainings (**E**–**H**), 200× magnifications, scalebars = 200 µm).

**Figure 8 membranes-12-00378-f008:**
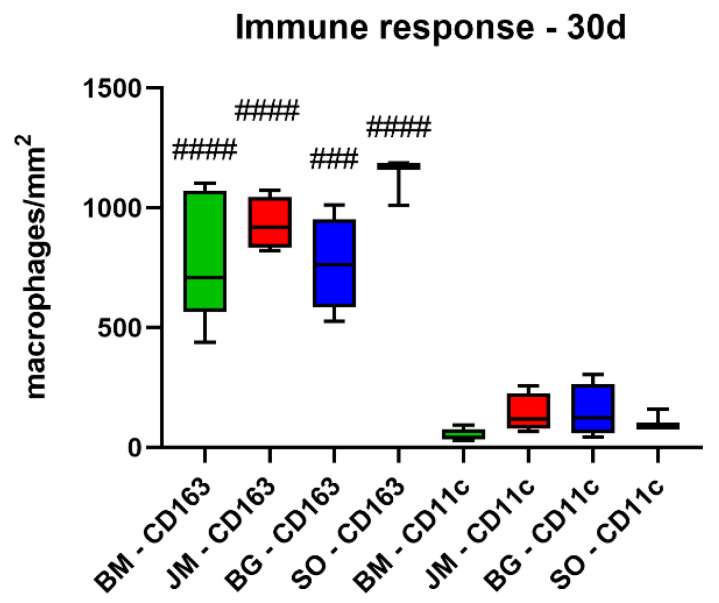
Results of the histomorphometrical measurements of the macrophage distribution at day 30 post implantationem (# = interindividual differences; ### *p* < 0.001, #### *p* < 0.0001).

**Figure 9 membranes-12-00378-f009:**
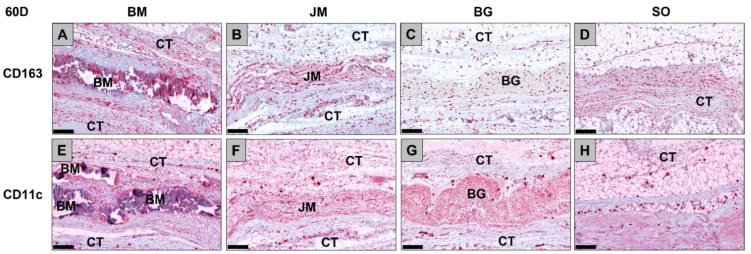
Exemplary histological images of the distribution of anti-inflammatory CD163-positive macrophages (**A**–**D**) and pro-inflammatory CD11c-positive macrophages (**E**–**H**) (red staining) within the implant beds of the three membranes within the subcutaneous connective tissue (CT) at day 60 post implantationem. (**A**) BM = Bovine membrane, (**B**) JM = Jason^®^ membrane, (**C**) BG = Bio-Gide^®^ membrane, (**D**) Sham operation group without biomaterial insertion (CD163-immunostainings (**A**–**D**) and CD11c-immunostainings (**E**–**H**), 200× magnifications, scalebars = 200 µm).

**Figure 10 membranes-12-00378-f010:**
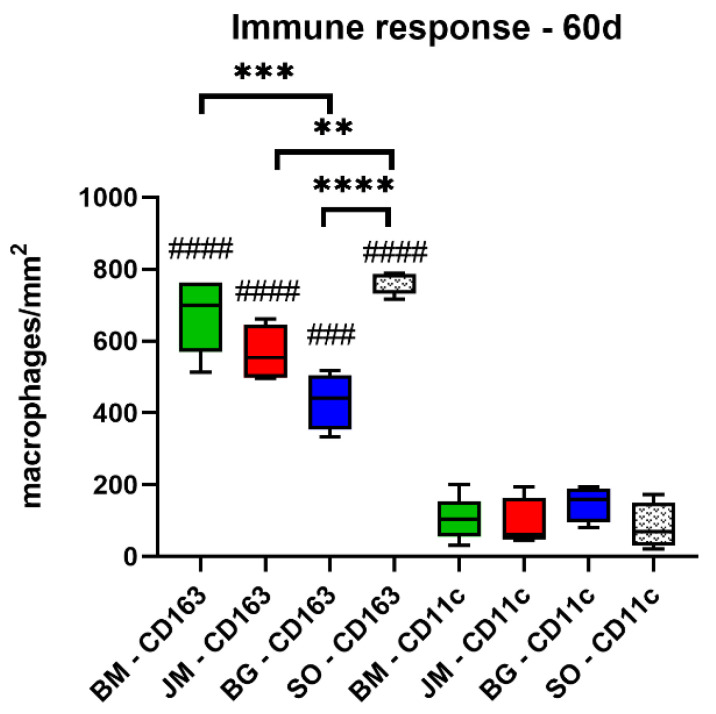
Results of the histomorphometrical measurements of the macrophage distribution at day 60 post implantationem (* = intraindividual differences, # = interindividual differences; ****/####: *p* < 0.0001, ***/###: *p* < 0.001 and **: *p* < 0.01). BM: bovine membrane, JM: Jason^®^ membrane, BG: Bio-Gide^®^, SO: sham operation.

**Figure 11 membranes-12-00378-f011:**
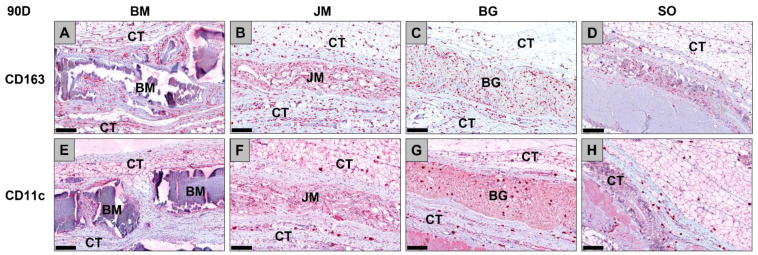
Exemplary histological images of the distribution of anti-inflammatory CD163-positive macrophages (**A**–**D**) and pro-inflammatory CD11c-positive macrophages (**E**–**H**) (red staining) within the implant beds of the three membranes within the subcutaneous connective tissue (CT) at day 90 post implantationem. (**A**) BM = Bovine membrane, (**B**) JM = Jason^®^ membrane, (**C**) BG = Bio-Gide^®^ membrane, (**D**) Sham operation group without biomaterial insertion (CD163-immunostainings (**A**–**D**) and CD11c-immunostainings (**E**–**H**), 200× magnifications, scalebars = 200 µm).

**Figure 12 membranes-12-00378-f012:**
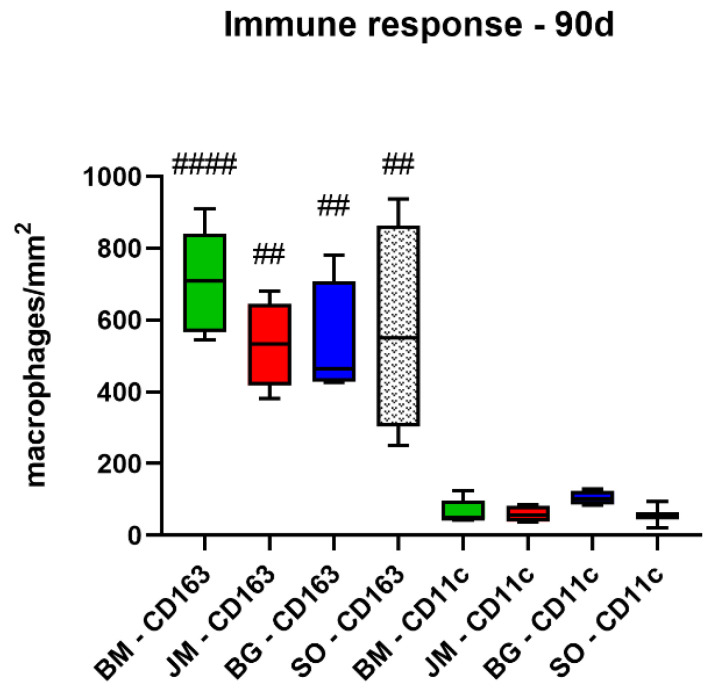
Results of the histomorphometrical measurements of the macrophage distribution at day 90 post implantationem (# = interindividual differences; ##: *p* < 0.01 and #### *p* < 0.0001). BM: bovine membrane, JM: Jason^®^ membrane, BG: Bio-Gide^®^, SO: sham operation.

**Figure 13 membranes-12-00378-f013:**
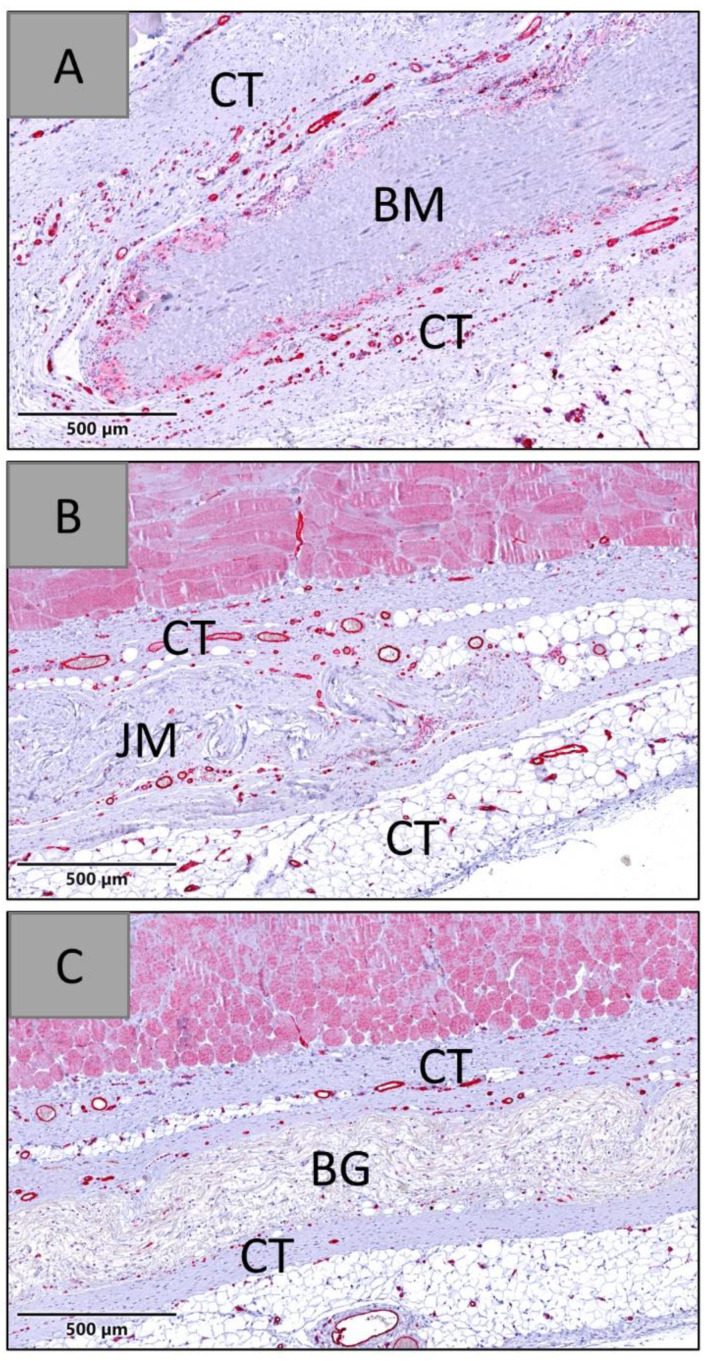
Exemplary histological images of the three membranes within the subcutaneous connective tissue (CT) at day 30 post implantationem. (**A**) BM = native bovine membrane, (**B**) JM = Jason^®^ membrane, (**C**) BG = Bio-Gide^®^, red staining = blood vessels (CD31-immunostainings, 20× magnifications, scalebars = 500 µm).

**Figure 14 membranes-12-00378-f014:**
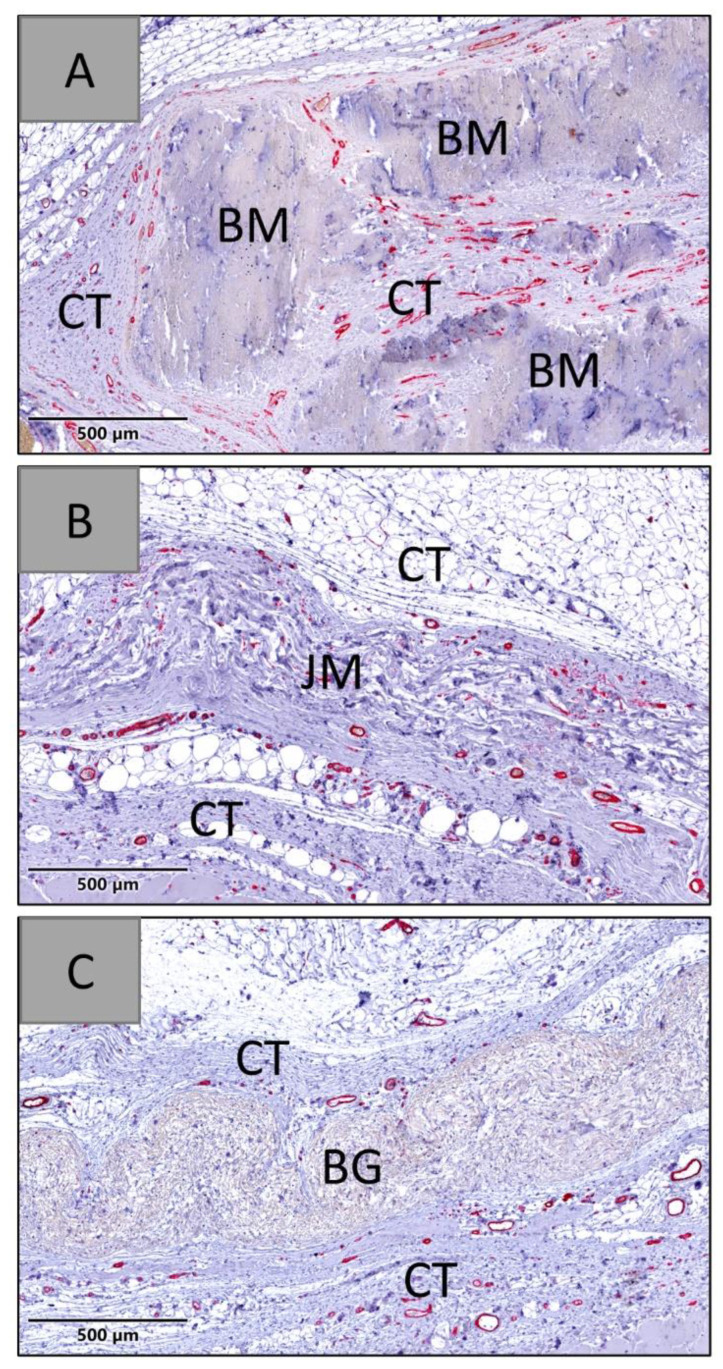
Exemplary histological images of the three membranes within the subcutaneous connective tissue (CT) at day 60 post implantationem. (**A**) BM = native bovine membrane, (**B**) JM = Jason^®^ membrane, (**C**) BG = Bio-Gide^®^ (CD31-immunostainings, 20× magnifications, scalebars = 500 µm).

**Figure 15 membranes-12-00378-f015:**
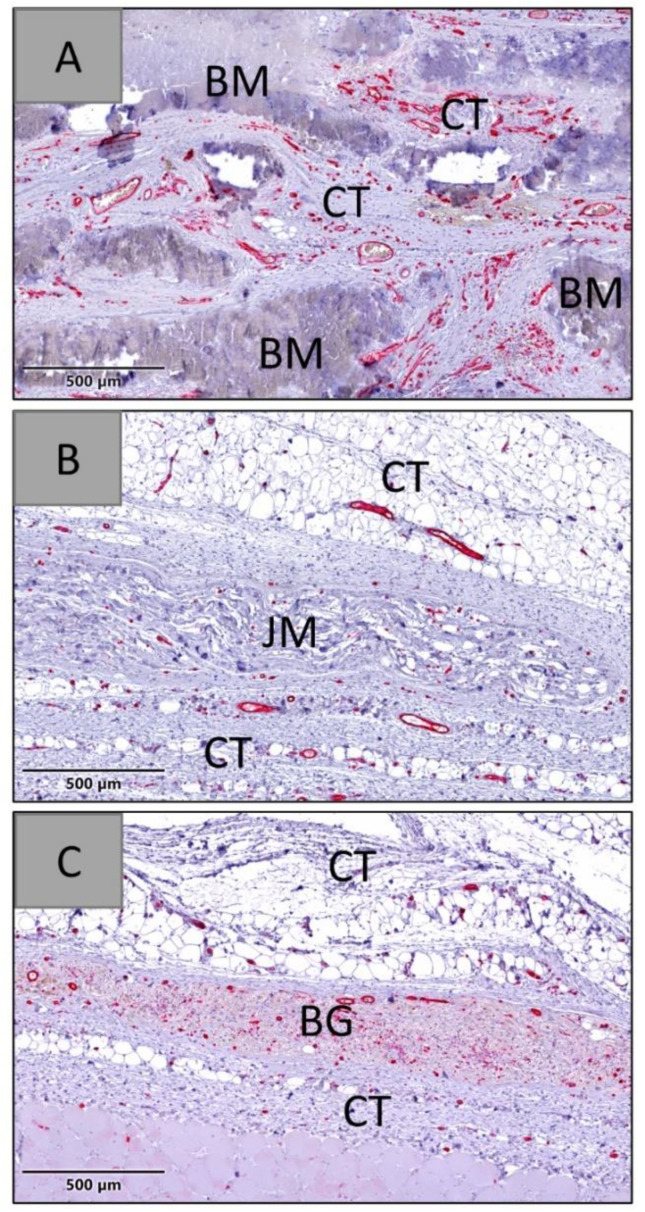
Exemplary histological images of the three membranes within the subcutaneous connective tissue (CT) at day 90 post implantationem. (**A**) BM = native bovine membrane, (**B**) JM = Jason^®^ membrane, (**C**) BG = Bio-Gide (CD31-immunostainings, 20× magnifications, scalebars = 500 µm).

**Figure 16 membranes-12-00378-f016:**
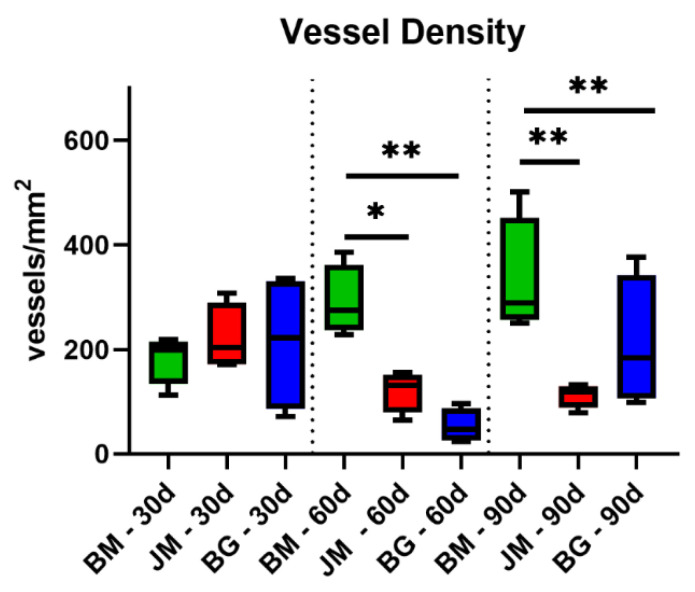
Results of the histomorphometrical measurements of the vessel density of the membrane areas (* = intraindividual differences; *: *p* < 0.05 and **: *p* < 0.01).

**Figure 17 membranes-12-00378-f017:**
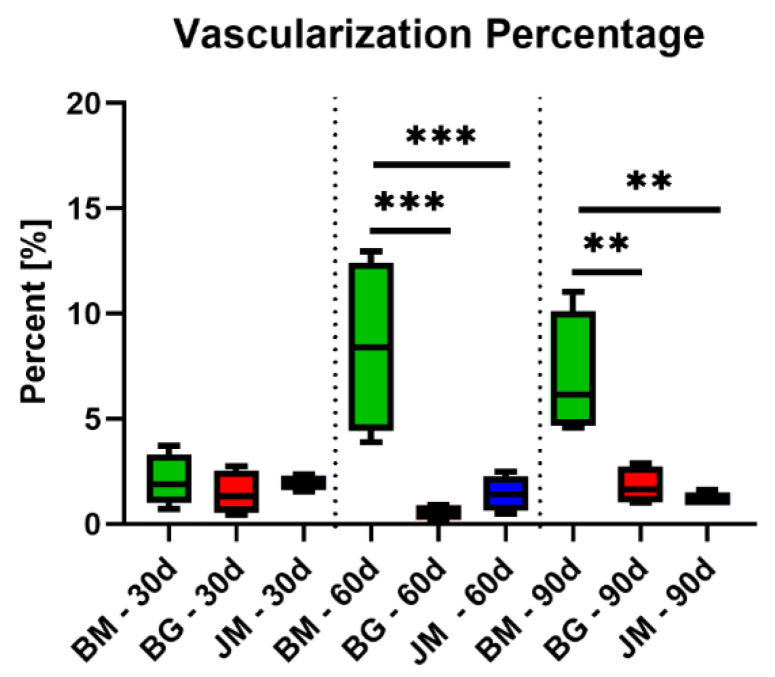
Results of the histomorphometrical measurements of the percent vascularization of the membrane areas (* = intraindividual differences; **: *p* < 0.01 and ***: *p* < 0.0001). BM: bovine membrane, JM: Jason^®^ membrane, BG: Bio-Gide^®^.

**Figure 18 membranes-12-00378-f018:**
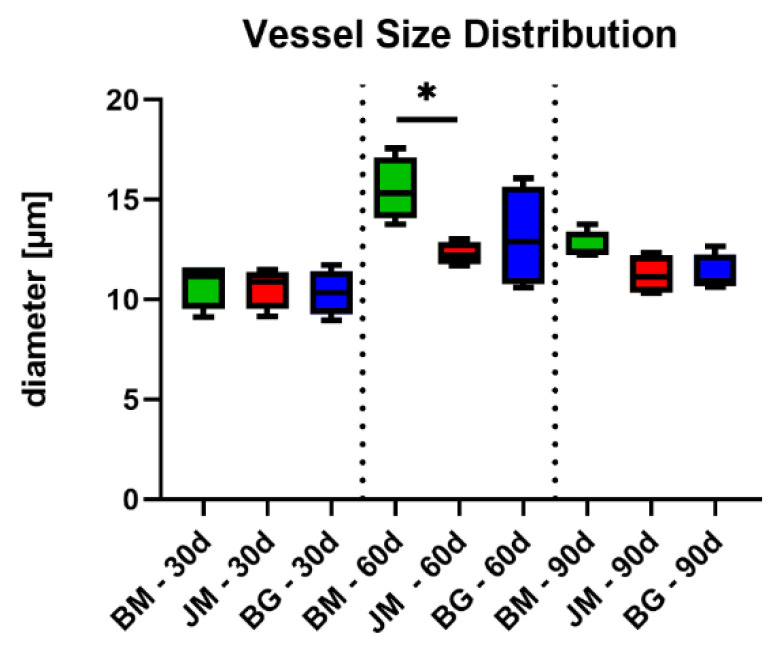
Results of the histomorphometrical measurements of the vessel sizes within the membrane areas (* = intraindividual differences; *: *p* < 0.05). BM: bovine membrane, JM: Jason^®^ membrane, BG: Bio-Gide^®^.

**Table 1 membranes-12-00378-t001:** Test and Control Article Descriptions.

Function	Name	Description	Implant Sizes (mm × mm)
Test Article	Bovine membrane	Native dermis-based Collagen membrane	10.0 × 10.0
Positive Control 1	Bio-Gide^®^ membrane	Native dermis-based collagen membrane	10.0 × 10.0
Positive Control 2	Jason^®^ membrane	Native pericardium-based collagen membrane	10.0 × 10.0
Negative Control	Sham operation	without biomaterial insertion	-

**Table 2 membranes-12-00378-t002:** Histologic Evaluation System for Irritancy/Reactivity—Cell Type/Response.

Response	Score (phf = Per High Powered (×400) Field)				
	0	1	2	3	4
Polymorphonuclear cells	0	Rare, 1–5/phf	6–10/phf	Heavy infiltrate	Packed
Lymphocytes	0	Rare, 1–5/phf	6–10/phf	Heavy infiltrate	Packed
Plasma cells	0	Rare, 1–5/phf	6–10/phf	Heavy infiltrate	Packed
Macrophages	0	Rare, 1–5/phf	6–10/phf	Heavy infiltrate	Packed
Giant cells	0	Rare, 1–2/phf	3–5/phf	Heavy infiltrate	Packed
Necrosis	0	Minimal	Mild	Moderate	Marked
Neovascularization	0	Minimal capillary proliferation focal, 1–3 buds	Groups of 4–7 capillaries with supporting fibroblastic structures	Broad band of capillaries with supporting structures	Extensive band of capillaries with supporting fibroblastic structures
Fibrocytes/fibroconnective tissue, fibrosis	0	Narrow band	Moderately thick band	Thick band	Extensive band
Fatty infiltrate	0	Minimal amount of fat associated with fibrosis	Several layers of fat and fibrosis	Elongated and broad accumulation of fat cells about the implant site	Extensive fat completely surrounding the implant
Irritancy score = (Polymorphonuclear Cells + Lymphocytes + Plasma Cells + Macrophages + Giant Cells + Necrosis) × 2 + (Neovascularization + Fibrosis + Fatty Infiltrate).					

**Table 3 membranes-12-00378-t003:** Irritancy/Reactivity Grade. Adapted from DIN EN ISO 10993-6.

Overall Irritancy Score	Irritancy/Reactivity Status
0.0 to 2.9	Minimal or no reaction (non-irritant)
3.0 to 8.9	Slight reaction (slight irritant)
9.0 to 15.0	Moderate reaction (moderate irritant)
>15.1	Severe reaction (severe irritant)

**Table 4 membranes-12-00378-t004:** Results of the biomaterial scoring evaluations at 10 days post implantationem. BM: bovine-based membrane, BG: Bio-Gide^®^ membrane, JM: Jason^®^ membrane, and SO: sham operation.

Parameter	Mean ± SD—Inflammation and Inflammatory Cell Types at Day 10			
	BM	JM	BG	SO
Polymorphonuclear Cells	1.00 ± 0.8	0.30 ± 0.1	0.40 ± 0.1	0.38 ± 0.1
Lymphocytes	0.75 ± 0.3	0.40 ± 0.1	0.70 ± 0.3	0.63 ± 0.3
Plasma Cells	0.00 ± 0.0	0.00 ± 0.0	0.00 ± 0.0	0.21 ± 0.1
Macrophages	2.00 ± 0.0	1.50 ± 0.4	1.80 ± 0.3	1.58 ± 0.4
Giant Cells	0.67 ± 0.3	0.40 ± 0.2	0.25 ± 0.2	0.00 ± 0.0
Neovascularization	0.67 ± 0.3	0.90 ± 0.2	0.60 ± 0.2	1.08 ± 0.4
Fibrosis	0.00 ± 0.0	0.10 ± 0.2	0.10 ± 0.2	0.00 ± 0.0
Fatty infiltrate	0.00 ± 0.0	0.00 ± 0.0	0.00 ± 0.0	0.00 ± 0.0
Necrosis	0.00 ± 0.0	0.00 ± 0.0	0.00 ± 0.0	0.00 ± 0.0

**Table 5 membranes-12-00378-t005:** Results of the biomaterial scoring evaluation at 30 days post implantationem. BM: bovine-based membrane, BG: Bio-Gide^®^ membrane, JM: Jason^®^ membrane, and SO: sham operation.

Parameter	Mean ± SD—Inflammation and Inflammatory Cell Types at Day 30			
	BM	JM	BG	SO
Polymorphonuclear Cells	0.29 ± 0.1	0.29 ± 0.2	0.50 ± 0.3	0.45 ± 0.1
Lymphocytes	0.96 ± 0.7	1.00 ± 0.5	0.67 ± 0.5	0.60 ± 0.5
Plasma Cells	0.00 ± 0.0	0.00 ± 0.0	0.00 ± 0.0	0.1 ± 0.1
Macrophages	1.83 ± 0.3	2.08 ± 0.2	2.00 ± 0.7	1.80 ± 0.3
Giant Cells	0.58 ± 0.3	0.13 ± 0.1	0.58 ± 0.6	0.05 ± 0.1
Neovascularization	0.08 ± 0.1	0.17 ± 0.1	0.20 ± 0.2	1.30 ± 0.4
Fibrosis	0.00 ± 0.0	0.04 ± 0.1	0.20 ±0.2	0.00 ± 0.0
Fatty infiltrate	0.00 ± 0.0	0.00 ± 0.0	0.08 ± 0.1	0.55 ± 0.3
Necrosis	0.04 ± 0.1	0.04 ± 0.1	0.00 ± 0.0	0.00 ± 0.0

**Table 6 membranes-12-00378-t006:** Results of the biomaterial scoring evaluation at 60 days post implantationem. BM: bovine-based membrane, BG: Bio-Gide^®^ membrane, JM: Jason^®^ membrane, and SO: sham operation.

Parameter	Mean ± SD—Inflammation and Inflammatory Cell Types at Day 60			
	BM	JM	BG	SO
Polymorphonuclear Cells	0.25 ± 0.2	0.75 ± 0.5	0.50 ± 0.4	0.67 ± 0.2
Lymphocytes	0.58 ± 0.5	0.54 ± 0.2	0.33 ± 0.1	0.50 ± 0.2
Plasma Cells	0.00 ± 0.0	0.00 ± 0.0	0.00 ± 0.0	0.0 ± 0.0
Macrophages	2.08 ± 0.4	1.50 ± 0.4	1.08 ± 0.2	1.33 ± 0.4
Giant Cells	0.67 ± 0.5	0.08 ± 0.1	0.08 ± 0.1	0.00 ± 0.0
Neovascularization	0.42 ± 0.1	0.08 ± 0.1	0.08 ± 0.1	0.83 ± 0.4
Fibrosis	0.13 ± 0.1	0.00 ± 0.0	0.13 ± 0.14	0.00 ± 0.0
Fatty infiltrate	0.17 ± 0.1	0.00 ± 0.0	0.04 ± 0.1	0.33 ± 0.4
Necrosis	0.00 ± 0.0	0.00 ± 0.0	0.00 ± 0.0	0.00 ± 0.0

**Table 7 membranes-12-00378-t007:** Results of the biomaterial scoring evaluation at 90 days post implantationem. BM: bovine-based membrane, BG: Bio-Gide^®^ membrane, JM: Jason^®^ membrane, and SO: sham operation.

Parameter	Mean ± SD—Inflammation and Inflammatory Cell Types at Day 90			
	BM	JM	BG	SO
Polymorphonuclear Cells	0.33 ± 0.1	0.75 ± 0.4	0.96 ± 0.3	0.33 ± 0.1
Lymphocytes	0.83 ± 0.4	0.41 ± 0.3	0.54 ± 0.3	0.33 ± 0.1
Plasma Cells	0.00 ± 0.0	0.00 ± 0.0	0.00 ± 0.0	0.0 ± 0.0
Macrophages	1.92 ± 0.2	1.17 ± 0.3	1.08 ± 0.2	0.54 ± 0.3
Giant Cells	0.83 ± 0.4	0.08 ± 0.1	0.29 ± 0.2	0.00 ± 0.0
Neovascularization	0.58 ±0.2	0.13 ± 0.1	0.08 ± 0.1	0.46 ± 0.2
Fibrosis	0.00 ± 0.0	0.00 ± 0.0	0.17 ± 0.1	0.00 ± 0.0
Fatty infiltrate	0.08 ± 0.1	0.17 ± 0.1	0.17 ± 0.1	0.42 ± 0.1
Necrosis	0.00 ± 0.0	0.00 ± 0.0	0.00 ± 0.0	0.00 ± 0.0

**Table 8 membranes-12-00378-t008:** Irritancy scores and irritancy status of the bovine-derived membrane at 10-, 30-, 60-, and 90-days post implantationem.

	Study Group	Treatment Irritancy Score	Overall Irritancy Score	Irritant Status
Day 10	BM	9.50	2.90	Non-irritant
	JM	6.20	6.60	-
	BG	7.00		-
	SO	6.67	-	-
Day 30	BM	7.50	0.0	Non-irritant
	JM	7.29	7.61	-
	BG	7.92		-
	SO	7.85	-	-
Day 60	BM	7.88	2.84	Non-irritant
	JM	5.83	5.04	-
	BG	4.25		-
	SO	6.17	-	-
Day 90	BM	8.50	2.85	Non-irritant
	JM	5.12	5.65	-
	BG	6.17		-
	SO	3.29	-	-

**Table 9 membranes-12-00378-t009:** Results of the Thickness Measurements.

Membrane/Time Point	Thickness (mm)			
	Day 10	Day 30	Day 60	Day 90
BM	0.54 ± 0.15	0.77 ± 0.22	0.59 ± 0.18	0.36 ± 0.13
JM	0.20 ± 0.04	0.28 ± 0.07	0.31 ± 0.05	0.26 ± 0.11
BG	0.45 ± 0.10	0.26 ± 0.09	0.27 ± 0.08	0.22 ± 0.05

**Table 10 membranes-12-00378-t010:** Results of the Macrophage Measurements/Immune Responses.

Membrane/Time Point	Day 10	Day 30	Day 60	Day 90
CD163 (cells/mm^2^)				
BM	848.9 ± 26.7	797.4 ± 273.4	673.0 ± 105.9	704.2 ± 146.3
JM	970.9 ± 300.4	933.3 ± 110.0	567.9 ± 75.3	531.8 ± 121.7
BG	1303.0 ± 592.1	766.5 ± 199.1	423.7 ± 78.2	534.2 ± 166.5
SO	708.7 ± 65.6	1123.0 ± 99.3	766.6 ± 34.0	572.3 ± 290.4
CD11c (cells/mm^2^)				
BM	173.3 ± 56.2	52.3 ± 26.2	104.3 ± 62.2	65.8 ± 34.7
JM	137.7 ± 65.6	140.3 ± 82.0	90.5 ± 70.0	59.1 ± 23.2
BG	382.2 ± 204.3	148.6 ± 111.0	148.5 ± 50.0	104.3 ± 20.4
SO	162.7 ± 91.5	109.6 ± 42.0	82.8 ± 64.2	56.3 ± 36.6

**Table 11 membranes-12-00378-t011:** Histomorphometrical results of the membrane vascularization. (BM: native bovine membrane, JM: Jason^®^ membrane, BG; Bio-Gide^®^ membrane).

	Day 30	Day 60	Day 90
BM	Vessel Density (vessels/mm^2^)		
	182.6 ± 47.73	291.0 ± 67.80	332.30 ± 114.70
	Vessel Percentage (%)		
	2.055 ± 1.246	8.405 ± 4.169	6.978 ± 2.955
	Vessel Diameter (µm)		
	10.78 ± 1.179	15.51 ± 1.587	12.67 ± 0.7209
JM	Vessel Density (vessels/mm^2^)		
	221.4 ± 64.56	120.70 ± 39.53	112.20 ± 23.15
	Vessel Percentage (%)		
	1.978 ± 0.365	1.448 ± 0.8435	1.163 ± 0.3278
	Vessel Diameter (µm)		
	10.61 ± 1.019	12.29 ± 0.5697	11.23 ± 1.008
BG	Vessel Density (vessels/mm^2^)		
	212.9 ± 131.7	53.52 ± 32.52	211.00 ± 125.80
	Vessel Percentage (%)		
	1.455 ± 1.065	0.5575 ± 0.4081	1.803 ± 0.9028
	Vessel Diameter (µm)		
	10.34 ± 1.065	13.13 ± 2.585	11.28 ± 0.9466

## Data Availability

All data are included in the manuscript.

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
