# Peer review of "In Vivo Biocompatibility Analysis of a Novel Barrier Membrane Based on Bovine Dermis-Derived Collagen for Guided Bone Regeneration (GBR)"

_membranes, 2022, doi:10.3390/membranes12040378_

Round 1

Reviewer 1 Report

Review comments of membranes -1658222 manuscript

The current manuscript entitled “In Vivo Biocompatibility Analysis of a Novel Barrier Mem brane Based on Bovine Dermis-derived Collagen for Guided Bone Regeneration”. Authors have demonstrated the results with adequate experimental evidence and the manuscript was written with scientific soundings, so I would recommend this manuscript for publication without further revision and I would be concerned with the following comment.

  1. In the manuscript, it would be better to show the schematic representation of collagen based membrane development for bone regeneration.

Author Response

Dear colleague many thanks for the nice comments.

Based on your idea we included a vschematic representation of the development of medical devices.

Best wishes

Mike Barbeck

Reviewer 2 Report

This is a comprehensive and well-written paper. The authors presented the work in a systematic way, and utilizing two control systems along with 4 time-points allow for good credibility for their findings.

There weren't many flaws I could find with it, but if I had to nitpick there are just a few:

  • Some very minor grammar/format issues here and there. (Eg. L97-L98 - such a -> such as)
  • The authors call it a 'novel barrier membrane' and while they presented in much detail the in vivo characterizations and analyses, I feel the process of synthesis/preparation itself could have used a bit more explanation. Hence, minor revision is required. 
  • The authors indicated that that the fragmentation of the BM membrane after 'Day 90' does not hinder the barrier functionality because of the special 'roof tile-like arrangement' degradation pattern. I think an image of this 'roof tile-like' fragmentation would have been great.

Author Response

Dear colleague many thanks for the nice comments.

We revised the manuscript based on your comments.

  • Some very minor grammar/format issues here and there. (Eg. L97-L98 - such a -> such as)

We corrected the errors now.

  • The authors call it a 'novel barrier membrane' and while they presented in much detail the in vivo characterizations and analyses, I feel the process of synthesis/preparation itself could have used a bit more explanation. Hence, minor revision is required. 

Dear colleague, we cannot include more information as we did not get it from the manufacturer. However, we hope that the manuscript is nevertheless suitable for publication.

  • The authors indicated that that the fragmentation of the BM membrane after 'Day 90' does not hinder the barrier functionality because of the special 'roof tile-like arrangement' degradation pattern. I think an image of this 'roof tile-like' fragmentation would have been great.

Dear colleague, the requested image is already integrated in Figure 2M.

Best wishes

Mike Barbeck